# Research on the Current Control Strategy of a Brushless DC Motor Utilizing Infinite Mixed Sensitivity Norm

**Tianqing Yuan [1], Jiu Chang [2,\*] and Yupeng Zhang [3]**

[1] Key Laboratory of Modern Power System Simulation and Control & Renewable Energy Technology, Ministry of Education, Northeast Electric Power University, Jilin 132012, China; 20192929@neepu.edu.cn
[2] Department of Electrical Engineering, Northeast Electric Power University, Jilin 132012, China
[3] School of Information Science and Engineering, Northeastern University, Shenyang 110819, China; 2310263@stu.neu.edu.cn
\* Correspondence: 2202100007@neepu.edu.cn; Tel.: +86-157-65545764

**Abstract:** During the brushless DC (BLDC) motor working process, the system encounters inevitable uncertainties. These ambiguities stem from potential fluctuations, random occurrences, measurement inaccuracies, varying operational conditions, environmental shifts such as temperature alterations, among other factors. Uncertainties, an inherent aspect of any real control system, can be broadly classified into two categories: sensor signal uncertainties and discrepancies between the mathematical and actual models due to parameter perturbations. To mitigate the impact of sensor noise and parameter perturbations on the BLDC motor, a robust control strategy utilizing infinite norm mixed sensitivity based on PI control strategy (PI-H∞-MIX) is proposed in this paper. Firstly, the closed-loop control structure and transfer function model of the BLDC motor control system current loop are analyzed based on the current loop circuit topology, and then, the model parameters perturbation is analyzed, and the multiplicative uncertainty bound is given. In addition, the appropriate weighting function is selected to ensure the robustness of the system. In this case, the controller design problem is transformed into the H∞ standard control problem, and then, the system augmentation matrix is established, and the controller is solved by Matlab/Simulink. Finally, the performances of the traditional PI control strategy and the PI-H∞-MIX are compared and analyzed. The results show that (1) the proposed PI-H∞-MIX strategy can improve the control system robustness under the parameter perturbation condition effectively, and (2) the proposed PI-H∞-MIX strategy can suppress the noise signal of the sensor.

**Keywords:** mixed sensitivity; robustness; brushless DC motor; current control

## 1. Introduction

The brushless DC (BLDC) motor owns superior control performance, mainly manifested in good speed control performance, wide speed regulation range, large starting torque and high efficiency [1–4]. The BLDC motor is widely used in many high-precision industries, such as defense, aerospace, robotics, industrial control processes and other fields [5–7].

In the BLDC motor control system using a cascade control structure, a controller is usually adopted in the inner current loop to make sure the output parameters of the motor can meet the requirements of stable operation [8]. Another controller is usually adopted in the outer speed loop [9–11]. Additionally, the outer speed loop generally controls the speed through adjusting the voltage or pulse-width modulation (PWM) regulation. Thus, the BLDC motor operation performance is mainly determined by the inner current loop controller and the outer speed loop controller [12–14].

To further improve the stability of the BLDC motor and reduce the torque ripple, many scholars have carried out various in-depth research. In [15], a Zeta converter is used to control the speed of the BLDC motor; hence, the motor speed is relatively constant

under static or dynamic conditions. The inner current loop mainly adopts proportional integral differential (PID) control, and the literature [16] combines the basic theory of fuzzy control with traditional PID control to design a fuzzy automatic adjustment PID controller. Compared with a traditional PID control method, the fuzzy automatic adjustment controller has a faster response speed and higher accuracy. In [17], the BLDC motor was controlled through self-tuning PID parameters. This method aims to use the collected data of detected back electromotive force (EMF) and rotor position information as feedback for the control algorithm to achieve self-tuning control. In [18], a new three-effective vector (TEV) current control scheme was proposed, which overcomes the current distortion that may be caused by the reverse magnetic field of phase C, resulting in small output current ripple and a faster response speed. The robust control algorithm has also been applied in motor control. An improved infinite norm (H∞) strategy was proposed in [19], and the improved controller can reduce the torque ripple and improve the dynamic response performance.

The above control strategies can improve the operation performance of the BLDC motor partly, but the influence of parameter perturbation and sensor noise is ignored. Therefore, the change in the motor system working conditions will deteriorate the influence of the parameter perturbation. In order to improve the dynamic response performance of the BLDC motor control system and enhance the robustness of the uncertain system under the parameter perturbation condition, a current control method that includes PI control and the H∞ mixed sensitivity robust control algorithm is proposed in this paper to optimize the operation of the BLDC motor system. Firstly, according to the circuit topology and the differential equations of the BLDC motors, the current loop transfer functions of the BLDC motors are derived. Then, according to the possible changes during the whole operating process, an uncertainty model of the BLDC motors control system is established. In addition, the optimal weighting function is calculated based on the mixed sensitivity theory, and a generalized feedback system augmented matrix is established to solve the ideal controller. Finally, the effectiveness of the proposed algorithm is verified through comparing with traditional PI controller.

## 2. Mathematical Model of BLDC Motor

The three-phase BLDC motor system is shown in Figure 1. According to the logic signals of the Hall elements, the power devices $T_1 \sim T_6$ can control the conduction and shutdown of the currents in the three-phase windings of the motor. A, B and C correspond to the three phases of the BLDC motor. The working state of the two-phase conduction is selected to established a BLDC motor model.

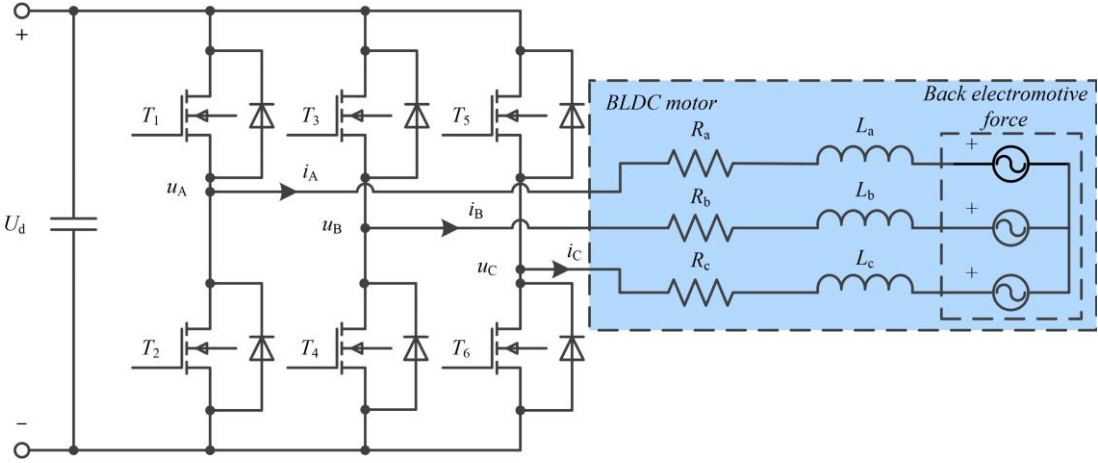

**Figure 1.** The circuit structure of the brushless DC motor.

According to Figure 1, the voltage equation matrix representation of the BLDC motor can be obtained:

$$\begin{bmatrix} u_A \\ u_B \\ u_C \end{bmatrix} = \begin{bmatrix} R & 0 & 0 \\ 0 & R & 0 \\ 0 & 0 & R \end{bmatrix} \begin{bmatrix} i_A \\ i_B \\ i_C \end{bmatrix} + \begin{bmatrix} L-M & 0 & 0 \\ 0 & L-M & 0 \\ 0 & 0 & L-M \end{bmatrix} \frac{d}{dt} \begin{bmatrix} i_A \\ i_B \\ i_C \end{bmatrix} + \begin{bmatrix} e_A \\ e_B \\ e_C \end{bmatrix} \tag{1}$$

where $u_A$, $u_B$ and $u_C$ are the three-phase voltages of A, B and C, respectively. $i_A$, $i_B$ and $i_C$ are the phase currents of A, B, and C, respectively. $R$ is the equivalent resistance in the three-phase symmetrical winding, satisfying $R = R_a = R_b = R_c$; $L$ is the equivalent self-inductance in the three-phase symmetrical winding, satisfying $L = L_a = L_b = L_c$; $M$ is the equivalent mutual inductance in the three-phase symmetrical winding. $e_A$, $e_B$, and $e_C$ are the back electromotive force of the three-phase winding, respectively.

As the BLDC motor is operated in 120° mode, the currents of the upper switch conducting phase and lower switch conducting phase are definied as equal in magnitude and opposite in direction. The instantaneous upper and lower switch conducting phases are defined as phase A and phase B, respectively. The conducting phases currents are expressed as:

$$\begin{cases} i_A = -i_B = i \\ \frac{di_A}{dt} = -\frac{di_B}{dt} = \frac{di}{dt} \end{cases} \tag{2}$$

where $i$ is the phase current of the winding in steady state.

From Equations (1) and (2), the conductive phases voltage equation can be obtained and expressed as:

$$u_{AB} = U_d = 2Ri + 2(L-M)\frac{di}{dt} + 2e_A = r_a i + L_0 \frac{di}{dt} + k_e \omega_0 \tag{3}$$

where $U_d$ is the DC voltage, $r_a$ is the winding line resistance, satisfying $r_a = 2R$; $L_0$ is the winding equivalent line inductance, satisfying $L_0 = 2(L - M)$; $k_e$ is the line back-potential coefficient, satisfying $k_e = 2p\psi_m$; $p$ is the pole logarithm of the motor; $\psi_m$ is the maximum value of the permanent magnet flux linkage in each phase winding; $\omega_0$ is mechanical angular velocity of the motor.

The equivalent armature loop can be obtained according to Equation (3), as shown in Figure 2.

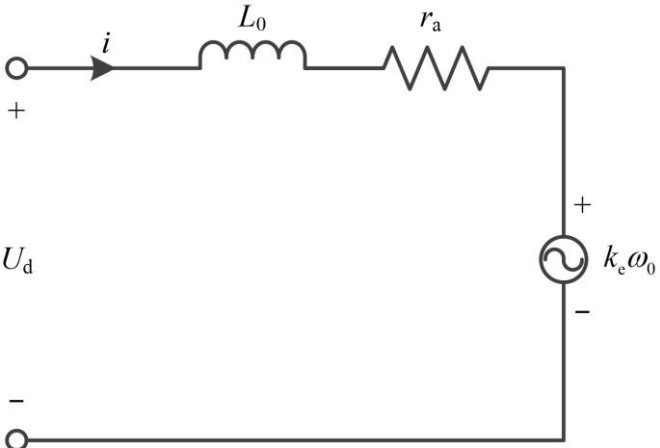

**Figure 2.** The equivalent circuit of the BLDC motor.

The electromagnetic torque of the motor is denoted as:

$$T_e = \frac{e_A + e_B + e_C}{\omega} = 2p\psi_m i_A = K_T i \tag{4}$$

where $K_T$ is the motor torque coefficient.

To establish a complete mathematical model of an electromechanical system, the motion equation of the motor is also needed, which can be given as:

$$T_e - T_L = K_T i - T_L = J\frac{d\omega}{dt} + B_v\omega \tag{5}$$

where $T_L$ is the load torque, $J$ is the rotor rotational inertia, and $B_v$ is the viscous friction coefficient.

The steady-state current $i$ can be written by transforming Equation (5):

$$i = \frac{J}{K_T}\frac{d\omega}{dt} + \frac{B_v}{K_T}\omega \tag{6}$$

And the voltage equation can also be rewritten by substituting Equation (6) into (3):

$$U_d = r_a\left(\frac{J}{K_T}\frac{d\omega}{dt} + \frac{B_v}{K_T}\omega\right) + L_0\frac{d}{dt}\left(\frac{J}{K_T}\frac{d\omega}{dt} + \frac{B_v}{K_T}\omega\right) + k_e\omega \tag{7}$$

Thus, the transfer function of the BLDC motor can be obtained as:

$$G(s) = \frac{K_T}{L_0Js^2 + (r_aJ + L_0B_v)s + (r_aB_v + k_eK_T)} \tag{8}$$

Based on above derivation, the system structure diagram of the BLDC motor can be established, as shown in Figure 3.

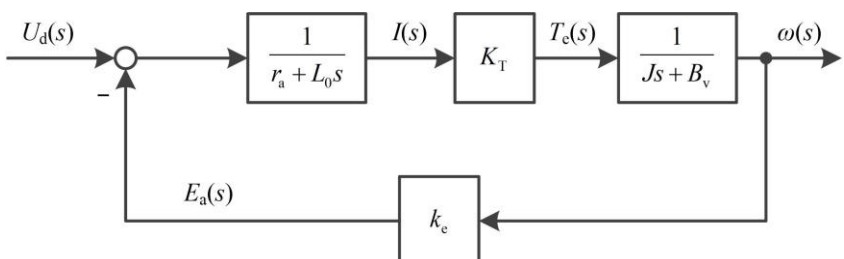

**Figure 3.** Structure diagram of the BLDC motor system.

The mechanical time constant $t_m$ and the electromagnetic time constant $t_e$ are described as:

$$\begin{cases} t_m = \frac{r_aJ + L_0B_v}{r_aB_v + k_eK_T} \\ t_e = \frac{L_0J}{r_aJ + L_0B_v} \end{cases} \tag{9}$$

Thus, Equation (8) can be rewritten as:

$$G(s) \approx \frac{K_T}{r_aB_v + k_eK_T} \cdot \frac{1}{s^2t_mt_e + st_m + st_e + 1} = \frac{K_T}{r_aB_v + k_eK_T} \cdot \frac{1}{(st_m + 1)(st_e + 1)} \tag{10}$$

According to Equation (10), the BLDC motor can be expressed by two series inertial elements. Since the compensator studied in this paper is used in the inner current loop of the control system, the equivalent system structure diagram of the BLDC motor can be finally determined based on Equation (10) and Figure 3, as shown in Figure 4.

$$G_0(s) = \frac{G_c(s) \cdot \frac{K_T}{r_aB_v + k_eK_T} \cdot \frac{1}{st_e+1}}{1 + G_c(s) \cdot \frac{K_T}{r_aB_v + k_eK_T} \cdot \frac{1}{st_e+1}} \tag{11}$$

where $G_c(s)$ is the PI link, satisfying $G_c(s) = (k_p s + k_i)/s$, and $I_{ref}$ is the reference current. According to Figure 4, the transfer function of BLDC motor is $G_0(s)$, which can be expressed as Equation (11).

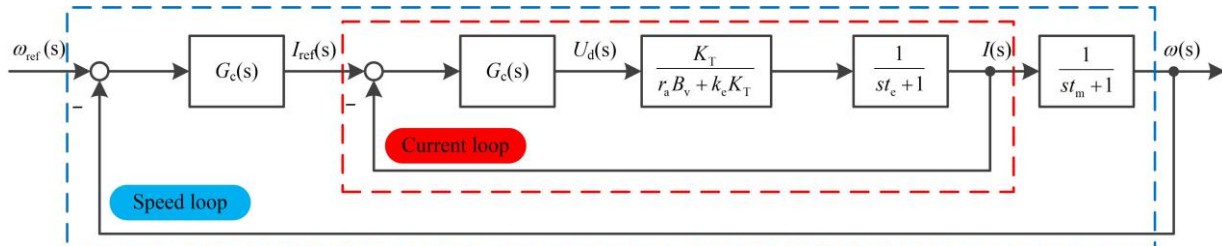

**Figure 4.** The double closed loop system of the BLDC motor.

## 3. H∞ Mixed Sensitivity Current Control

The theory of the H∞ mixed sensitivity belongs to the robust control theories, the main content of the robustness analysis of the BLDC motor control system is how to design the robust controller for the control object, and it can be divided into three aspects as follows:

(1)    The design method of the controller;
(2)    The sufficient and necessary conditions for the existence of the controller;
(3)    Analysis, design and implementation.

Almost all the control problems can be described in Figure 5, the generalized control object contains the set of the actual control object, the actuator, sensors, etc. The aim of this paper is to make the H∞ mixed sensitivity control system equivalent to the general control system, as shown in Figure 5.

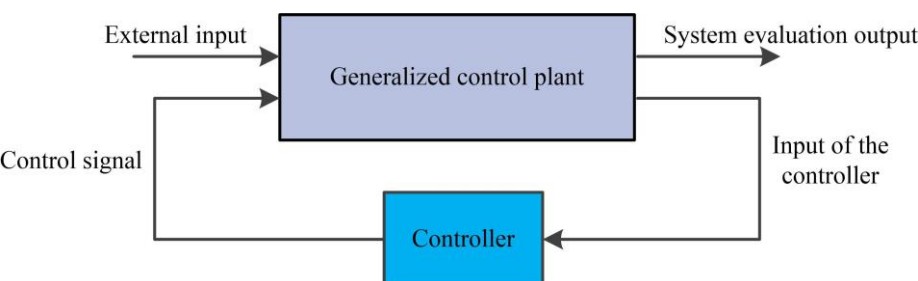

**Figure 5.** The general control system.

### 3.1. Uncertainty Analysis of the System Parameters

During the BLDC motor working process, the motor parameters may vary caused by uncertain factors [20]. Thus, the nominal model will change in the designing of the controller due to the parameter perturbations. Additionally, the resistance and inductance perturbations are the primary factors [21–24]. Therefore, the uncertainty of the resistance and the inductance should be mainly considered during the establishing process of the actual motor model. Since the influences of the resistance uncertainty and the inductance uncertainty are similar, to simplify the analysis process, this paper only considers the influence of the resistance parameter perturbation on the system, and the parameter perturbation range of resistance is set to [−20%, 20%]. The set nominal parameters of the motor are shown in Table 1.

**Table 1.** Nominal parameters of the BLDC motor.

| Parameter | Value |
| --- | --- |
| Resistance (Ω) | 0.6 |
| Inductance (H) | 0.002 |
| Mutual Inductance (H) | 0.001 |
| Rotational Inertia of the motor (Kg·m$^2$) | 0.00005 |
| Hysteresis Coefficient | 0 |
| Motor Torque Coefficient | 0.015 |
| Motor Back Electromotive Force Coefficient | 0.6 |

The relationship between the nominal system $G_0(s)$ and the actual system $P(s)$ is:

$$P(s) = [1 + \Delta W(s)]G_0(s) \tag{12}$$

In Equation (12), $\Delta$ represents an uncertain perturbation module that satisfies $\|\Delta\|_\infty < 1$, $W(s)$ is the system transfer function of multiplicative uncertainty.

The transfer function $W(s)$ of the multiplicative uncertainty bound satisfies:

$$\left| \frac{P(j\omega)}{G_0(j\omega)} - 1 \right| \le |W(j\omega)|, \forall R \in [0.48, 0.72] \tag{13}$$

Through substituting the nominal model parameters and the set parameter perturbation bounds into Equation (13), the parameter perturbation bound function can be obtained as:

$$W(s) = \frac{0.87s^2}{4.14s^2 + 3100s + 50,000} \tag{14}$$

The block diagram of a current loop closed-loop system with multiplicative uncertainty is shown in Figure 6.

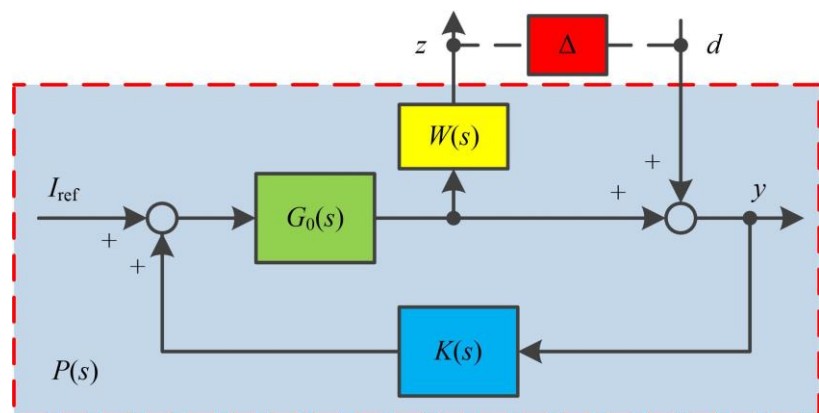

**Figure 6.** Block diagram of the multiplicative uncertainty system.

For the control system with multiplicative uncertainty, as shown in Figure 6, the actual control object $P(s)$ belonging to the unstructured can be set as:

$$U_M = \left\{ P(s) = [I + W(s)\Delta(s)]G_0(s), \Delta(s) \in \overline{BH_\infty} \right\} \tag{15}$$

The goal of the robust stabilization control for $U_M$ is to find a controller $K(s)$, which can make the feedback control system stable for any $\Delta(s) \in BH_\infty$.

From Figure 6, the transfer relationship from $d$ to $z$ can be obtained as:

$$z = -G_0(s)K(s)[z + W(s)d] \tag{16}$$

It can also be written as:

$$z = -[I + G_0(s)K(s)]^{-1}G_0(s)K(s)W(s)d \tag{17}$$

It can be seen that for any $\Delta(s) \in BH_\infty$, the sufficient and necessary conditions for the stability of the closed-loop control system is:

$$\left\| (I + G_0(s)K(s))^{-1}G_0(s)K(s)W(s) \right\|_\infty \le 1 \tag{18}$$

It can also be written as:

$$(I + G_0(s)K(s))^{-1}G_0(s)K(s)W(s) \in BH_\infty \qquad (19)$$

### 3.2. Analysis of H∞ Mixed Sensitivity Control Problem

The mixed sensitivity problem can also be seen as the infinite norm problem. The uncertainty of the model should be considered in the system using mixed sensitivity algorithms, and the interference of external factors should also be suppressed [25–28]. Therefore, the uncertainty of the model and the suppression performance of the external factors' interference should be considered in the weighting functions selecting process. The design of an H∞ norm-based controller is achieved through a generalized control object and the selection of appropriate weighting functions; the standard model for the mixed sensitivity problem is shown in Figure 7.

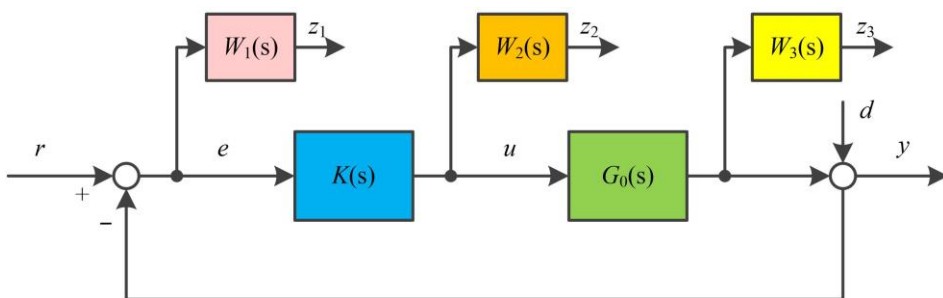

**Figure 7.** Mixed sensitivity problem model.

In Figure 7, $z_1$, $z_2$ and $z_3$ are the evaluation signals of the system, $r$ is the reference input, $e$ is the tracking error, $u$ is the control signal, $d$ is the current sensor noise, $y$ is the output signal, $K(s)$ is the controller to be solved, and $G_0(s)$ is the controlled object under nominal conditions.

It should be noted that the transfer function from disturbance $d$ to output $y$ is the sensitivity function $S(s)$, which can be calculated as:

$$S(s) = \frac{1}{1 + G_0(s)K(s)} \qquad (20)$$

At the same time, $S(s)$ is also the transfer function from reference input $r$ to tracking error $e$. The sensitivity function represents the ability of the closed-loop system to suppress disturbances [25]. Therefore, it can be seen that the disturbance can be suppressed through reducing the sensitivity function gain, however, the lower gain will weaken the control system robustness.

The transfer function from the reference input $r$ to the output $y$ is the complementary sensitivity function $T(s)$.

$$T(s) = \frac{G_0(s)K(s)}{1 + G_0(s)K(s)} \qquad (21)$$

The complementary sensitivity function $T(s)$ represents the size of the perturbation $\Delta$ in the system multiplicative perturbation $(I + \Delta)G_0$, which can describe the high-frequency unmodeled dynamic characteristics of the system. The stability robustness of the system can be ensured through introducing this function [25]. Since the sensitivity function and the complementary sensitivity function satisfy $S(s) + T(s) = I(s)$, the goals that improving the system's anti-interference ability and improving the stability robustness are contradictory, which should be determined by the actual motor operation condition.

To describe the mid-low frequency parameter perturbations of the system, the transfer function $R(s)$ is adopted and defined as:

$$R(s) = \frac{K(s)}{1 + G_0(s)K(s)} \tag{22}$$

The transfer function represents the magnitude of the perturbation $\Delta G_0$ in the additive perturbation $G_0 + \Delta G_0$ to the system, which is generally set to a constant.

It is also shown in Figure 7 that $W_1(s)$ is the weighting function of the sensitivity function $S(s)$, and $W_1(s)$ is also the performance constant of the control system. It should be noted that the influence of disturbances can be suppressed effectively through adjusting weighting function; thus, the desired output signal can be provided. $W_2(s)$ is the weighting function of $R(s)$, which can reflect the restriction on additive uncertainty, and the weighting function $W_2(s)$ can also be regarded as the constraint of the control signal amplitude. $W_3(s)$ is the weighting function of complementary sensitivity function $T(s)$, and $W_3(s)$ reflects the restriction on the multiplicative uncertainty, which is determined by the characteristics of the control plant.

The design of the mixed sensitivity controller involves selecting various weighting functions appropriately; thus, a positive and rational controller can be obtained. Hence, the closed-loop stability of the control system can be ensured, and the following relationship can be met.

$$\left\| \begin{matrix} W_1(s)S(s) \\ W_2(s)R(s) \\ W_3(s)T(s) \end{matrix} \right\|_\infty \leq \mu \tag{23}$$

where $\mu$ is the system performance indicator, as the performance indicator in the infinite norm theory, and a smaller value $\mu$ indicates that the impact of interference on the system error will be reduced to a minimum, and usually set to 1.

### 3.3. The Noise Signal of Sensor

In Figure 7, the interference signal $d$ and the reference signal $r$ serve as inputs to the mixed sensitivity problem model. Within the motor's control system, sensor noise bifurcates into two categories: Hall sensor noise and current sensor noise. The Hall sensor, influenced by alterations in magnetic induction intensity, generates a Hall voltage at its terminals. This signal, post transmission amplification, is relayed to the motor. Following data processing, it translates into the motor's position angle.

Consequently, the Hall sensor noise, equivalent to $d_n$, impacts the external speed loop. The current sensor operates on Faraday's law of electromagnetic induction to detect the current signal, rendering its noise equivalent to $d$, which influences the internal current loop as depicted in Figure 7.

### 3.4. Selection of Weighting Functions

The gain value of $W_1(s)$ should be sufficiently large in the low-frequency band; thus, the influence of the interference can be suppressed effectively and the input signal can be tracked accurately, and there are no strict requirements in the high-frequency band beyond the system requirements. Hence, the weighting function $W_1(s)$ of the sensitivity function $S(s)$ should meet the following condition.

$$W_1(s) = \frac{211}{s + 1000} \tag{24}$$

The weighted function $W_2(s)$ is used to limit the value of the control variable $u$. Hence, the control variable $u$ can remain in the system's allowable range and prevent the severe saturation of the control system during the whole working process. Therefore, the static gain

of $W_2(s)$ should be small enough in the context of the control system has sufficient bandwidth. The weighted function $W_2(s)$ of function $R(s)$ should meet the following condition.

$$W_2(s) = 0.008 \tag{25}$$

To improve the high-frequency band characteristics accuracy of the nominal plant and avoid the uncertainty of the gain and the phase of the control plant caused by unmodeled dynamic characteristics, a high-pass filter is selected as the weighted function $W_3(s)$ in this section. For the closed-loop control system shown in Figure 6, when $\Delta(s)$ is 0, the complement sensitivity function on the output side of the control object is shown in Equation (21). In this case, Equations (18) and (19) can be written as:

$$\|T(s)W(s)\|_\infty \leq 1 \tag{26}$$

$$T(s)W(s) \in BH_\infty \tag{27}$$

Therefore, the necessary and sufficient conditions can be converted into weighted constraints on the complementary sensitivity function for the robust stabilization of multiplicative uncertainty systems.

Due to $W_3(s)$ representing the norm bound of multiplicative perturbation, it can be seen from Equations (26) and (27) that $W_3(s)$ satisfies the design conditions of the mixed sensitivity controller that was described in Equation (23). Therefore, the weighted function $W_3(s)$ of the complementary sensitivity function $T(s)$ selected in this section should satisfy the following condition.

$$W_3(s) = W(s) = \frac{0.87s^2}{4.14s^2 + 3100s + 50,000} \tag{28}$$

*3.5. Calculation of Controller*

In order to meet the performance metrics in Equation (23) function, the system's input and output are reset and an augmented controlled system is established. The relevant vectors of the augmented controlled system are defined as:

$$\begin{cases} r = i_{\text{ref}} \\ z = \begin{bmatrix} z_1 & z_2 & z_3 \end{bmatrix}^{\text{T}} \\ y = e = i_{\text{ref}} - i \end{cases} \tag{29}$$

where $r$ represents the external input, $z$ represents the system evaluation output, and $y$ represents the input of the controller.

The block diagram of the generalized current loop feedback control system is shown in Figure 8.

It should be noted that, $P(s)$ in the generalized current loop feedback control system is the generalized controlled plant; the mathematical model of the $S/R/T$ mixed sensitivity problem containing the weighted functions can be evaluated by:

$$\begin{bmatrix} z_1 \\ z_2 \\ z_3 \\ y \end{bmatrix} = \begin{bmatrix} W_1(s) & -W_1(s)G_0(s) \\ 0 & W_2(s) \\ 0 & W_3(s) \\ I & -G_0(s) \end{bmatrix} \begin{bmatrix} r \\ u \end{bmatrix} = P \begin{bmatrix} r \\ u \end{bmatrix} = \begin{bmatrix} P_{11} & P_{12} \\ P_{21} & P_{22} \end{bmatrix} \begin{bmatrix} r \\ u \end{bmatrix} \tag{30}$$

where $I$ is the identity matrix.

Finally, the closed-loop transfer function from *r* to *z* can be obtained and expressed as:

$$T_{\mathrm{rz}}(s) = P_{11}(s) + P_{12}(s)K(s)[I - P_{22}(s)K(s)]^{-1}P_{21}(s) = \begin{bmatrix} W_1(s)S(s) \\ W_2(s)R(s) \\ W_3(s)T(s) \end{bmatrix} \tag{31}$$

Through substituting the data in Table 1, and the transfer functions in Equations (24), (25) and (28) into Equation (31), the controller can be determined through the augw function in Matlab as follows:

$$\begin{cases} K(s) = \begin{bmatrix} K_{11}(s) & K_{12}(s) \\ K_{21}(s) & K_{22}(s) \end{bmatrix} = \begin{bmatrix} -1000 & 0 & 0 & 0 & 0 & 1.916 \\ 0 & -16.49 & -109.9 & 55.7 & 2.533 & 0 \\ 0 & 0 & -732.3 & 371.1 & 16.88 & 0 \\ 0 & 0 & 0 & -16.57 & 21.99 & 0 \\ 9.83e4 & -179.1 & -1.767e4 & -1.244e5 & -5902 & 0 \\ 5.13e4 & -93.47 & -9221 & -6.457e4 & -2776 & 0 \end{bmatrix} \\ \mu = 0.1598 \end{cases} \tag{32}$$

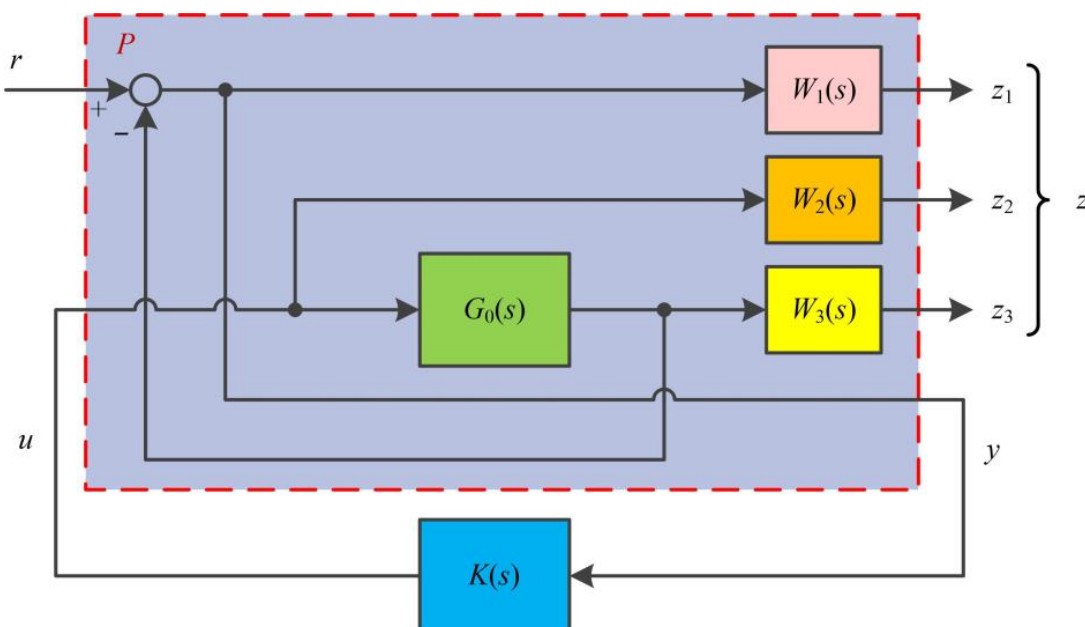

**Figure 8.** Block diagram of the generalized current loop feedback control system.

## 4. Simulation Analysis and Comparison

In order to verify the effectiveness of the proposed PI-H∞-MIX control strategy, the performance comparison between the PI-H∞-MIX control strategy and the traditional PI control strategy are carried out through Matlab/Simulink in this section. By optimizing the performance of the brushless DC motor, more suitable PI parameters can be determined with a value of 0.02 for $k_{\mathrm{p}}$ and 10 for $k_{\mathrm{i}}$ in PI link, and the BLDC motor parameters are shown in Table 1. The steady-state performance, the dynamic response performance and robustness under parameter perturbation of the BLDC motor driven by different control strategy are analyzed. The control system used the PI controller is defined as $S_{\mathrm{PI}}$, and the control system used the mixed sensitivity controller is defined as $S_{\mathrm{TIX}}$.

### 4.1. Steady-State Performance

The steady-state performance simulation of the BLDC motor driven by the traditional PI control strategy and the proposed PI-H∞-MIX control strategy are studied; the reference torque and speed are set to 0.5 N·m and 1000 rpm, respectively. The simulation results are

shown in Figure 9. It is obvious that the torque and speed simulation waveforms are nearly the same.

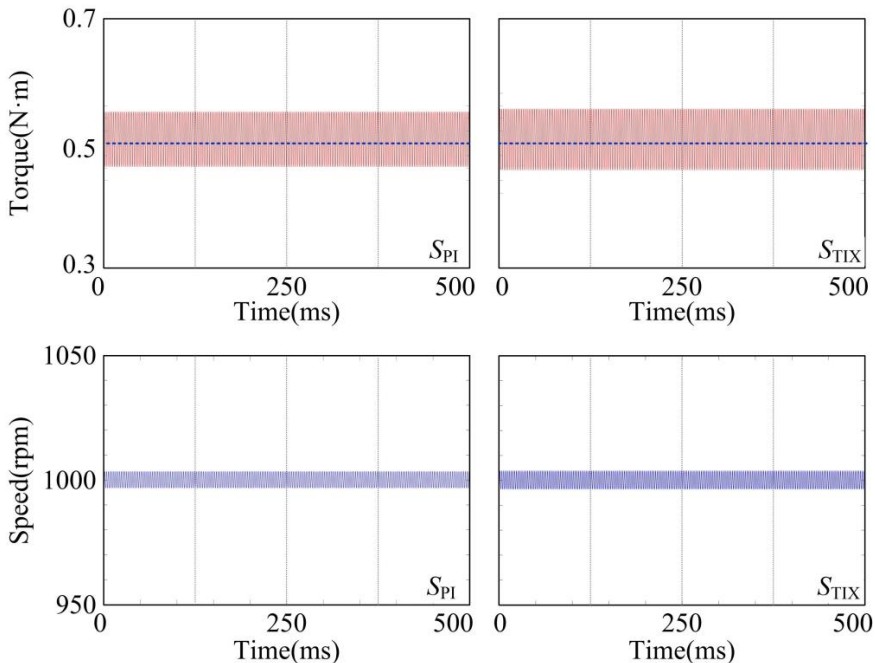

**Figure 9.** Steady-state performance waveforms of $S_{PI}$ and $S_{TIX}$.

*4.2. Dynamic Response Analysis*

In the dynamic simulation section, the reference speed is set to 1200 rpm, the load torque of the motor is changed from 0.3 N·m to 0.5 N·m and was restored back to 0.2 after 0.5 s. The simulation results are shown in Figure 10.

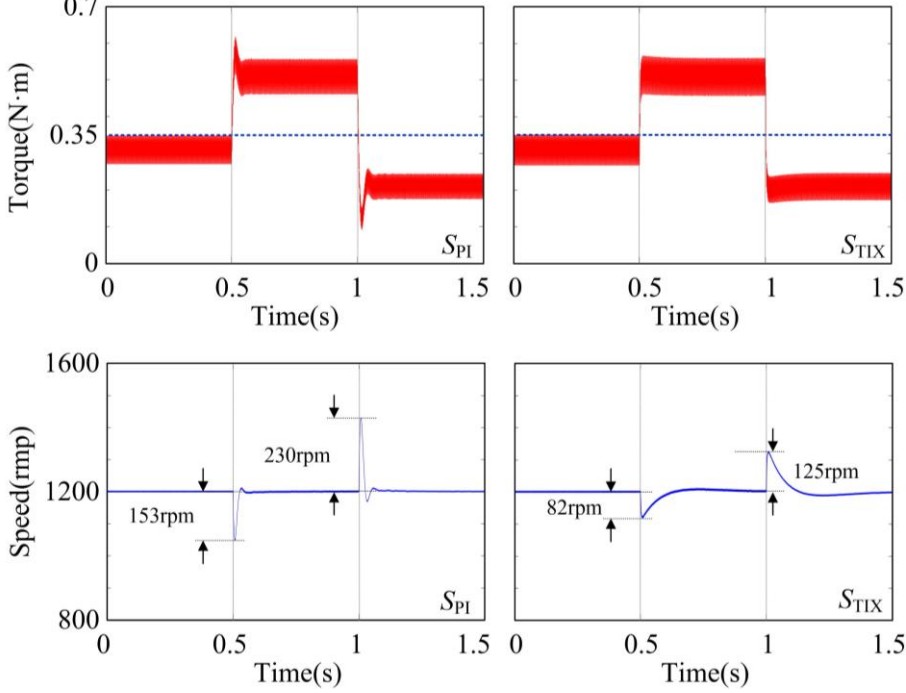

**Figure 10.** Dynamic respond performance waveforms of $S_{PI}$ and $S_{TIX}$ with the variation of the load torque.

When the motor load increased, the speed fluctuations of $S_{PI}$ and $S_{TIX}$ were 153 rpm and 82 rpm, respectively. When the motor load reduced, the speed fluctuations of $S_{PI}$ and $S_{TIX}$ were 230 rpm and 125 rpm, respectively.

From these results it can be found that when compared with a traditional PI control strategy, the PI-H∞-MIX control strategy can reduce the speed fluctuation by 46.4% when the motor load increased suddenly, and can also reduce the speed fluctuation by 45.6% when the motor load decreased suddenly.

In conclusion, the H∞ mixed sensitivity control strategy can suppress the dynamic speed overshoot of the motor control system while the load changed; thus, when the BLDC motor is running at the specified speed, the dynamic performance of the BLDC motor control system can be optimized.

### 4.3. Robustness Analysis under Parameter Perturbation

Due to the fluctuation of the resistance and the inductance being one of the main factors that generated the system's uncertainty, the robustness of the BLDC motor control system driven by the traditional PI control strategy and the proposed PI-H∞-MIX control strategy under the resistance perturbation and the inductance perturbation are analyzed in this section.

The simulation waveforms of $S_{PI}$ and $S_{TIX}$ when the resistance suddenly increases from 0.6 Ω to 1.2 Ω is shown in Figure 11.

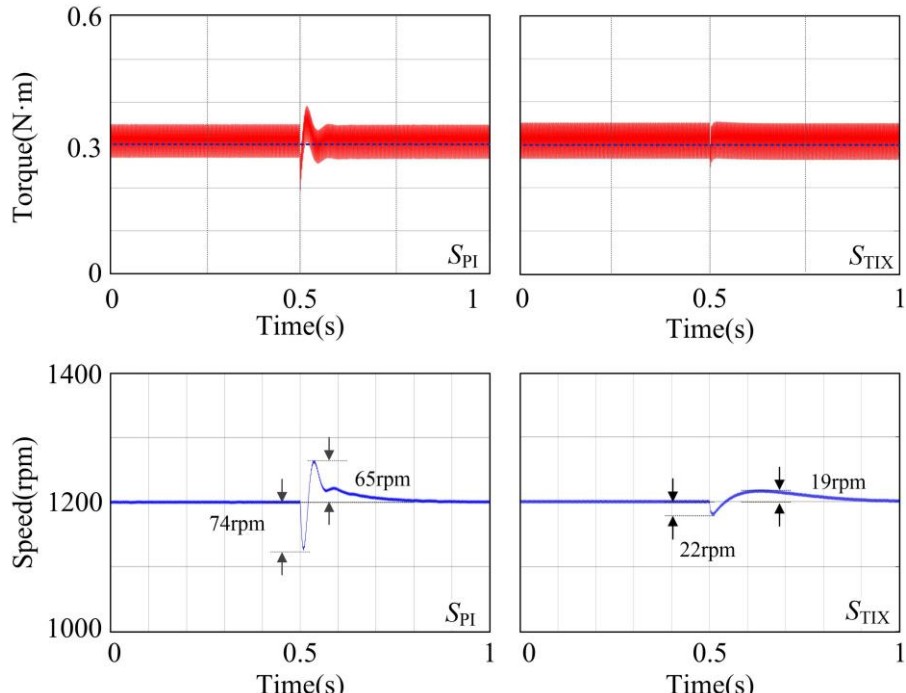

**Figure 11.** Dynamic respond performance waveforms of $S_{PI}$ and $S_{TIX}$ under the resistance perturbation.

When the resistance suddenly increases, the speed fluctuations of $S_{PI}$ and $S_{TIX}$ are 139 rpm and 41 rpm, respectively, and the torque ripple of $S_{TIX}$ is clearly lower than that of $S_{PI}$ while the resistance suddenly increases. We can thus understand that the PI-H∞-MIX control strategy can suppress the speed fluctuation by 70.5% than traditional PI control strategy. Thus, we then know that the PI-H∞-MIX control strategy can achieve the robustness of the motor control system under the resistance perturbation.

The simulation waveforms of $S_{PI}$ and $S_{TIX}$ when the inductance suddenly increases from 0.2 mH to 0.3 mH is shown in Figure 12.

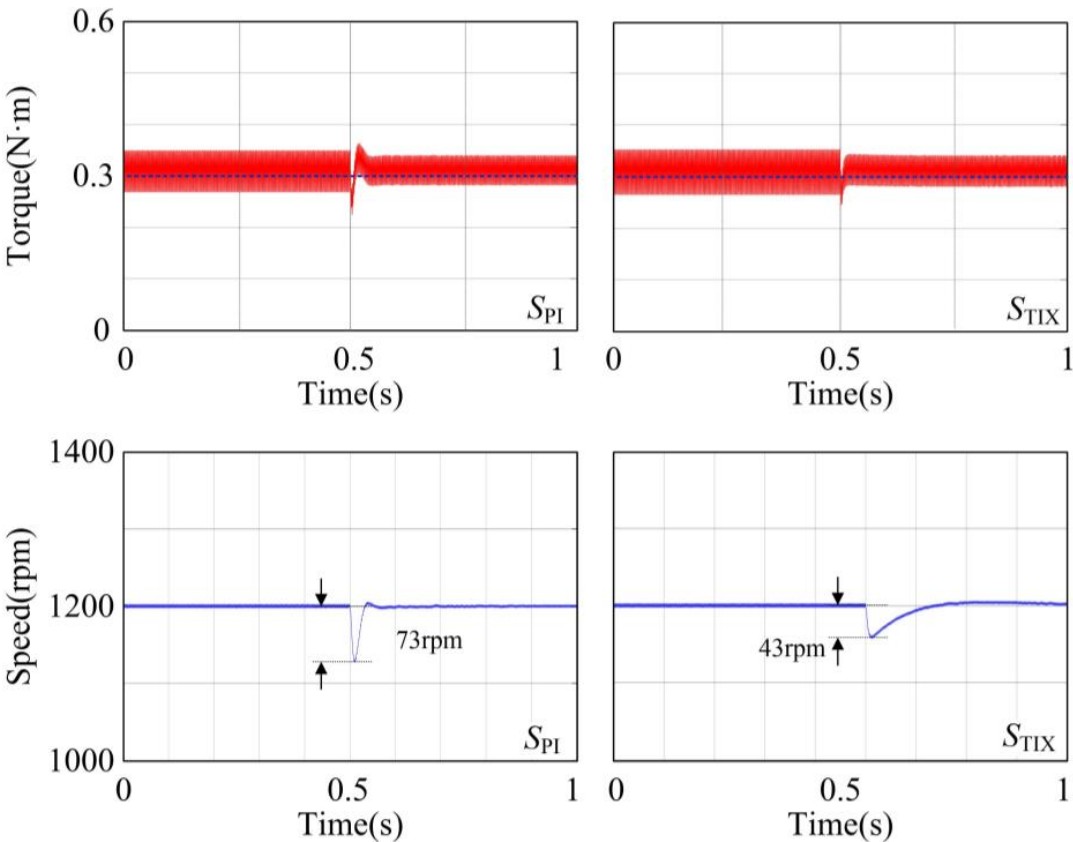

**Figure 12.** Dynamic respond performance waveforms of $S_{PI}$ and $S_{TIX}$ under the inductance perturbation.

When the inductance suddenly increases, the speed fluctuation of $S_{PI}$ and $S_{TIX}$ are 73 rpm and 43 rpm, respectively, and the torque ripple of $S_{TIX}$ is clearly lower than that of $S_{PI}$ while the resistance suddenly increases. We can thus understand that the PI-H∞-MIX control strategy can suppress the speed fluctuation by 70.5% than traditional PI control strategy. Thus, we then know that the PI-H∞-MIX control strategy can achieve the robustness of the motor control system under the inductance perturbation.

*4.4. Robustness Analysis under Sensor Noise*

The noise generated by the Hall sensor and the current sensor constitutes a significant source of system uncertainty. Noise, an ever-present factor, can be suppressed, but not entirely eradicated. This section analyzes the efficacy of noise suppression driven by the traditional PI control strategy and the proposed PI-H∞-MIX control strategy.

Initially, the motor is set to run smoothly at a speed of 900 rpm with a load of 0.3 N·m. At the 0.5 s mark, it is assumed that the Hall sensor receives an external noise interference of 0.01 s. At this point, the speed and torque graphs of $S_{PI}$ and $S_{TIX}$ are as shown in Figure 13. As shown in Figure 13, in terms of speed, the designed infinity norm compensator can suppress speed distortion amplitude by about 40%, and can recover to normal more quickly after the noise disappears. In terms of torque, the designed infinity norm compensator can also smooth torque fluctuations and has a faster response speed. Thus, the PI-H∞-MIX control strategy has good robustness under the influence of the Hall sensor noise.

Next, to verify the robustness of the proposed controller to the noise of the current sensor, when the motor is in a steady state, it is assumed that the current sensor receives an external noise interference of 0.01 s at 0.5 s. At this point, the speed and torque graphs of $S_{PI}$ and $S_{TIX}$ are as shown in Figure 14; thus, in terms of speed, the designed infinity norm compensator can suppress speed distortion amplitude by about 9.6%, and can recover to normal more quickly after the noise disappears. In terms of torque, the designed infinity

norm compensator can also smooth torque fluctuations and has a faster response speed. Therefore, the PI-H∞-MIX control strategy exhibits good control performance.

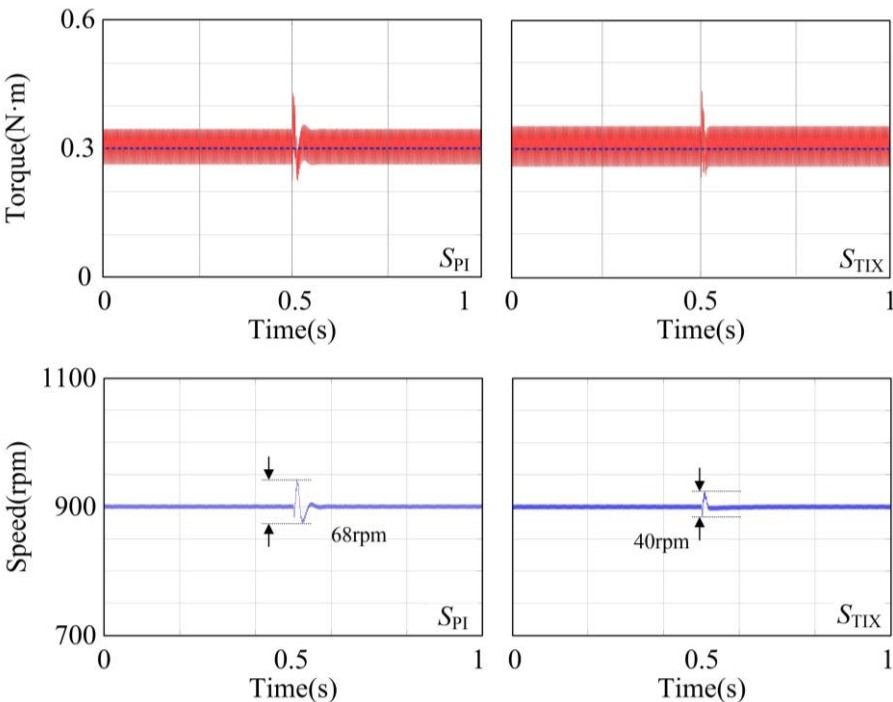

**Figure 13.** Dynamic respond performance waveforms of $S_{PI}$ and $S_{TIX}$ under the Hall sensor noise.

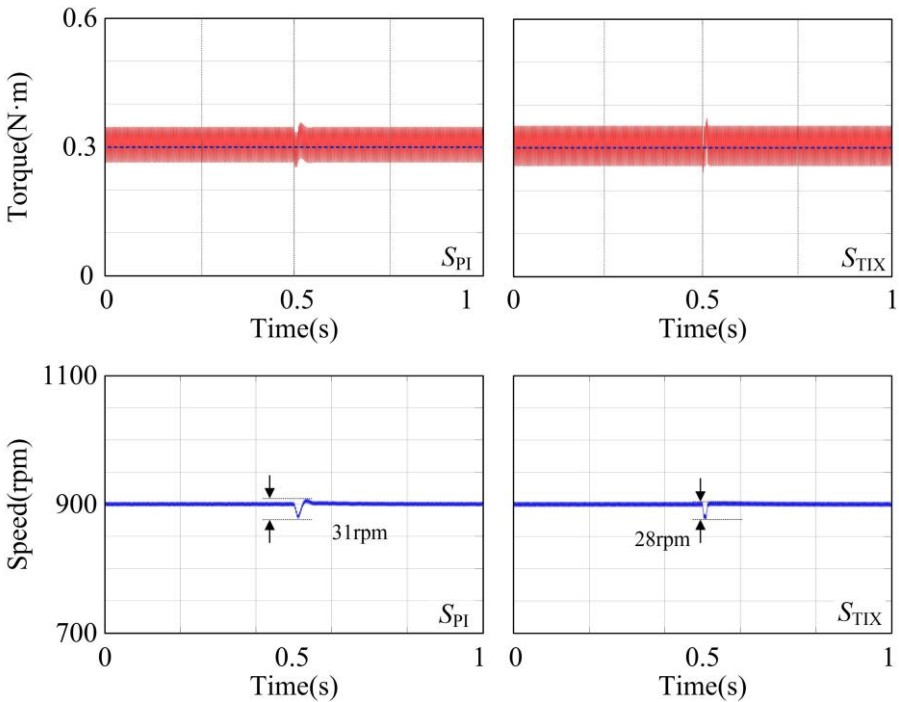

**Figure 14.** Dynamic respond performance waveforms of $S_{PI}$ and $S_{TIX}$ under the current sensor noise.

## 5. Discussion

In this paper, we focus on how to improve the dynamic performance and enhance the system robustness of BLDC motors. We categorize the above problem as a system uncertainty problem and obtain a current control compensator using an infinite-paradigm mixed-sensitivity robust control strategy. Some computational procedures and simulation setups will be added in this chapter.

In this paper, for the cases of load change, speed change and noise interference, we choose to derive the current loop transfer function in order to be able to describe these problems uniformly as the input and output problems of the infinite-paradigm standard control model. In the process of derivation, in order to simplify the calculation, we chose to take the two-phase conduction as the reference model, and complete the model for solving the infinite-paradigm standard problem. One concern about the model was that the commutation process is not considered. Because the commutation process is too complicated, there is no special simulation study in the simulation process. While this study did not conduct, theoretically, the internal structure change under the motor, the voltage change caused by the commutation should also be regarded as the internal interference of the system, which can be classified as the robustness problem of the system. In the simulation process, in order to enhance the universality of the simulation verification, this paper chooses the dual-loop motor control structure composed of speed ring and current ring, and empirical methods are adopted to adjust the PID parameters of the control system. Compared with some PID parameters obtained through the optimization algorithm, this approach is more realistic. The final solved controller is presented as a matrix, which can be programmed and written to the control board for control system optimization.

## 6. Conclusions

This paper proposes a novel PI-H∞-MIX control strategy for the BLDC motor, which can improve the dynamic performance of the control system and solve the poor robustness problem caused by parameter perturbation. The operation performance of the BLDC motor driven by a different control strategy is studied by simulation, and the simulation results clearly indicate that the PI-H∞-MIX control strategy owns the following advantages than the PI control strategy:

1. The PI-H∞-MIX control strategy can improve the dynamic response performance of the BLDC motor control system under load disturbance;
2. The robustness of the BLDC motor control system is greatly enhanced by the PI-H∞-MIX control strategy while considering the variation of the resistance parameter and the inductance parameter;
3. The PI-H∞-MIX control strategy can make the model of the control system more accurate by reducing the impact of noise interference on the system;
4. The PI-H∞-MIX control strategy can be used to solve the parameter fluctuations problems in grid-connected inverts and solve the load fluctuations in power grids, and other relevant fields.

**Author Contributions:** Conceptualization, T.Y., J.C. and Y.Z.; methodology, J.C.; software, J.C. and Y.Z.; validation, T.Y., J.C. and Y.Z.; formal analysis, T.Y., J.C. and Y.Z.; investigation, Y.Z.; resources, T.Y.; data curation, Y.Z.; writing—original draft preparation, J.C and Y.Z.; writing—review and editing, J.C. All authors have read and agreed to the published version of the manuscript.

**Funding:** This research was funded by the project is supported by Scientific Research Project of Education Department of Jilin Province (No. JJKH20230121KJ).

**Data Availability Statement:** Not applicable.

**Conflicts of Interest:** The authors declare no conflict of interest. The funders had no role in the design of the study; in the collection, analyses, or interpretation of data; in the writing of the manuscript; or in the decision to publish the results.

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
