# Peer review of "Research on the Current Control Strategy of a Brushless DC Motor Utilizing Infinite Mixed Sensitivity Norm"

_electronics, doi:10.3390/electronics12214525_

Round 1
Reviewer 1 Report
Comments and Suggestions for Authors
This article discusses the performance of the traditional PI control strategy and the PI-H∞-MIX for the BLDC motor control system, including comparisons and analysis.
I have some recommendations, notations, and questions.
Substantive errors:
Experimental results should be explained in more detail. I recommend adding a 'Discussion' chapter before the 'Conclusion.' For instance, you may consider elaborating on aspects such as how the PI controller parameters were obtained (the tuning method used), whether the system has been described sufficiently, whether any simplifications were made, and how the real implementation could be realized. Why only the current loop transfer function is derived in Chapter 2? Etc.
Line 24: Please provide the software version and provider name. Or you can add the homepage of the software.
Line 41: Abbreviation must be explained after the first occurrence.
Figure 1: Please add black dots where connections exist (to + or - poles, diodes, etc). The following three items on the right side (circles with tildes) are without names. The blue subsystem is the BLCD motor, not only BLCD! The name "BLDC" must be corrected to "BLDC motor". I recommend using the word principal electric wiring/circuit in the Figure description. There is no shown control system in this figure.
Line 85: A, B and C seem to be winding markings (labels), that need to be introduced.
Figure 4: The description should address not only the current loop but also the speed loop.
Line 128: There should be Gc(s), Gc is the function of the Laplace operator.
Line 130: Chapter 3: Format: Chapter without text. There should be some text between 3 and 3.1. Otherwise, chapter 3 is unnecessary.
Table 1 description: only motor, not control system!
Lines 363 and 364: "... the BLDC control ..." should be "... the BLDC motor control ..."
Formal mistakes:
Line 82: "Fig" without a dot "Fig.". I recommend using the complete word (Figure 1) instead of the short form in the whole article.
Line 84: uA, uB, uC must be written with indexes, such as equation uses. Big A, B and C must be introduced.
Equations 2, 3, 4, 5, etc.: there should be a comma before 'where' after 'equation,' or a period if it's the end of a sentence.
Equation 7: brackets size!
Line 146: The variables must be written in italic font. Please check this in the whole article. (See line 159, ...)
Lines 296-302: Please check the grammar.
Missing definite and indefinite articles, some sentences are incomplete, and some words are used incorrectly.
Author Response
Dear reviewer,
Thank you for checking our manuscript, we have made changes in this paper according to your suggestions. Please see the attachment.
Kind regards,
Jiu Chang
Northeast Electric Power University

Reviewer 2 Report
Comments and Suggestions for Authors
The paper presents the application of Hinfty controll to BLDC motor.
An important assumption made by authors is the simultaneous conduction of only 2 switches of the 2 level inverter. This simplifies the motor model to classical DC motor model.
The synthesis of Hinfty is known topic, its application to DC motor control is nice topic to visit. The paper is presented nicely.
The results suffer from the simulated and not experimental verification. This is mainly due to the assumtpion made on the 2 switches on controll method that can greately influence motor performance when coupled with a control algorithm. All the experimental assumptions such as "Hall sensor noise" should be avoided and used the actuall simulation event that happened (such as added noise to current feedback signal, etc..)
Please rename your section into Simulated analysis and comparison.
The comparison completed would stand better weight if it was done on some modern control method, rather than the PI control.
Furthermore, the PI controller tuning can be varied to get different speed/torque performance, you could change the PI response and get results much more simmilar to the Hinfty results you present. Study https://www.mdpi.com/2079-9292/11/10/1553 figures 7 and 10 for example on this. No need to include the citation in paper, just to be clear, its just to clarify my point for comparisons.
I believe that the simulation and comparison chapter is not sufficient to demonstrate that the Hinfty method is "better" in control terms for stated reasons.
Nevertheless, it is not reason to disregard the Hinfty as a control structure that can find its uses in the motor control. Therefore, I suggest to reformulate the text so it shows that you present Hinfty, not neccesarily that it is "better" because 2 questionable simulations show it that way. Just write the text like: "I made Hinfty work, and it works, best performance we achieved is this and this", do not make claims that it is better or worse than something else because you can not make that statement based on presented results. I find this problem in a lot of papers dealing with motor control, we have two approaches: "My speed/torque follows the reference, that means my control is good" and the "My performance is better then method X because it shows smaller signal dips on loading", none of these approaches actually mean something to a person waiting to use the presented methods in real controller design. If done experimentally, maybe.
Author Response

(The authors gave the same response as above.)

Reviewer 3 Report
Comments and Suggestions for Authors
To mitigate the impact of sensor nose and parameter perturbations on the BLDC motor, this paper proposed a robust control strategy utilizing infinite norm mixed sensitivity based on PI control strategy. My concerns are listed in the following.
- I suggest to use Figure instead of Fig to uniform the format.
- In Equation (13), why do you choose the range 0.48-0.72 for R?
- In line 225, please revise “Figures 6”.
- Please be careful about English writing, such as “is” in line 313 should be ‘are’.
Minor revision needed.
Author Response

(The authors gave the same response as above.)

Reviewer 4 Report
Comments and Suggestions for Authors
Good article to which I have a few comments.
1. It is not specified what parameters influence this particular selection of controller parameters and their limitations
2. Why weren't tests performed at one speed?
3. The effect of the regulators' actions on the disturbance is to reduce the amplitude of the electromagnetic Torque The consequence is the reduction of rotational speed ripples. Figures 9-13 should be shown as in Figure 8, i.e. first the torque, then the speed. The order of describing the influence of controllers on the electromagnetic torque and rotational speed should also be changed accordingly (and not the other way round).
4. The name of chapter 4 is misleading. It contains only the results of numerical calculations. The work does not contain any experimental research.
5. Analyzing an individual change in e.g. resistance or inductance has no practical significance. A change in resistance in a real system is associated with damage, e.g. a partial winding short circuit. This has a simultaneous effect on the inductance, voltage constant and torque constant. How will the proposed regulator respond to such a situation?
6. The value of the rotational inertia has quite a significant impact on the amplitude of the rotational speed. Is the rotational inertia given in Table 1 the rotational inertia of the rotor?
7. The proposed controller dampens electromagnetic torque oscillations more effectively, but is also slower. What it comes from?
Author Response

(The authors gave the same response as above.)

Round 2
Reviewer 1 Report
Comments and Suggestions for Authors
The article is edited according to the comments in a minimalistic form.
Author Response

(The authors gave the same response as above.)

Reviewer 2 Report
Comments and Suggestions for Authors
I believe that the authors missunderstood my previous review in terms of Comparison with PI control.
The authors letter clearly states that they beleive that the comparison should be made. And I want to be very clear that this is true and valid. Do not remove the comparison.
What you should modify is the claims that your method is better than PI, because with very simple re-tuning of the control gains it can be made other way around.
You have made comments about having X% better dynamic perfomance at lines 356 344 335 315. this is valid only for the presented 2 sets of tunings and therefor can not be claims for the Algorithm performance.
As you say in conclusion, the empirical tuning of PID is more realistic with regards to practical use of PID's, and that is the sole source of uncertanty with respect to performance metrics of the comparison. Thats why I beleive you should let the reader decide for himself what is "better", but dont remove the comparison, rather just say with this type of tuning we managed to get this results.
But deffinately comment on your tuning methodology and aimed performance goal that ultimately led to the PI parameters selection, so the reader has a benchmark for the PI.
Author Response

(The authors gave the same response as above.)
